# Prothrombinase-Induced Clotting Time to Measure Drug Concentrations of Rivaroxaban, Apixaban, and Edoxaban in Clinical Practice: A Cross-Sectional Study

**DOI:** 10.3390/life12071027

**Published:** 2022-07-11

**Authors:** Vepusha Sathanantham, Lorenzo Alberio, Cédric Bovet, Pierre Fontana, Bernhard Gerber, Lukas Graf, Adriana Mendez, Thomas C. Sauter, Adrian Schmidt, Jan-Dirk Studt, Walter A. Wuillemin, Michael Nagler

**Affiliations:** 1Department of Clinical Chemistry, Inselspital, Bern University Hospital, 3010 Bern, Switzerland; sathanantham@t-online.de (V.S.); cedric.bovet@protonmail.com (C.B.); 2Service and Central Laboratory of Hematology, CHUV, Lausanne University Hospital, 1011 Lausanne, Switzerland; lorenzo.alberio@chuv.ch; 3Division of Angiology and Hemostasis, Geneva University Hospital, 1205 Geneva, Switzerland; pierre.fontana@hcuge.ch; 4Clinic of Hematology, Oncology Institute of Southern Switzerland, 6500 Bellinzona, Switzerland; bernhard.gerber@eoc.ch; 5Faculty of Medicine, University of Zurich, 8091 Zurich, Switzerland; 6Centre for Laboratory Medicine St. Gallen, 9001 St. Gallen, Switzerland; lukas.graf@zlmsg.ch; 7Department of Laboratory Medicine, Cantonal Hospital Aarau, 5001 Aarau, Switzerland; adriana.mendez@ksa.ch; 8Department of Emergency Medicine, Inselspital, Bern University Hospital, 3010 Bern, Switzerland; thomas.sauter@insel.ch; 9Clinic of Medical Oncology and Hematology, Institute of Laboratory Medicine, City Hospital Waid and Triemli, 8063 Zurich, Switzerland; adrian.schmidt@triemli.zuerich.ch; 10Division of Medical Oncology and Hematology, University and University Hospital Zurich, 8091 Zurich, Switzerland; jan-dirk.studt@usz.ch; 11Division of Hematology and Central Hematology Laboratory, Cantonal Hospital of Lucerne, 6000 Lucerne, Switzerland; walter.wuillemin@luks.ch; 12Department of Clinical Chemistry, Inselspital, Bern University Hospital, University of Bern, 3010 Bern, Switzerland

**Keywords:** cross-sectional studies, diagnostic accuracy, sensitivity and specificity, prothrombinase-induced clotting time, laboratory monitoring, rivaroxaban, apixaban, edoxaban, direct oral anticoagulants

## Abstract

Prothrombinase-induced clotting time (PiCT) is proposed as a rapid and inexpensive laboratory test to measure direct oral anticoagulant (DOAC) drug levels. In a prospective, multicenter cross-sectional study, including 851 patients, we aimed to study the accuracy of PiCT in determining rivaroxaban, apixaban, and edoxaban drug concentrations and assessed whether clinically relevant drug levels could be predicted correctly. Citrated plasma samples were collected, and the Pefakit^®^ PiCT was utilized. Ultra-high performance liquid chromatography-tandem mass spectrometry (LC-MS/MS) was performed to measure drug concentrations. Cut-off levels were established using receiver-operating characteristics curves. We calculated sensitivities and specificities with respect to clinically relevant drug concentrations. Spearman’s correlation coefficient between PiCT and drug concentrations was 0.85 in the case of rivaroxaban (95% CI 0.82, 0.88), 0.66 for apixaban (95% CI 0.60, 0.71), and 0.78 for edoxaban (95% CI 0.65, 0.86). The sensitivity to detect clinically relevant drug concentrations was 85.1% in the case of 30 µg L^−1^ (95% CI 82.0, 87.7; specificity 77.9; 72.1, 82.7), 85.7% in the case of 50 µg L^−1^ (82.4, 88.4; specificity 77.3; 72.5, 81.5), and 85.1% in the case of 100 µg L^−1^ (80.9, 88.4; specificity 73.2%; 69.1, 76.9). In conclusion, the association of PiCT with DOAC concentrations was fair, and the majority of clinically relevant drug concentrations were correctly predicted.

## 1. Introduction

The proportion of individuals treated with direct oral anticoagulants (DOAC) is steadily increasing, because of their good manageability and a favorable risk–benefit ratio [1,2]. These patients are exposed to critical clinical situations, such as urgent surgery, major bleeding, and thrombolysis for acute stroke [3,4,5,6,7]. Current guidelines support the determination of DOAC drug levels in these situations, to decide upon reversal agents, scheduling of operations, and decisions for or against thrombolysis [3,7,8,9,10]. In addition, the drug accumulation in patients with renal impairment, hepatic failure, or old age can be detected [11,12,13,14]. Knowledge of DOAC drug concentrations is also important in the case of intoxication [3]. Some authors even argue that the monitoring of DOAC might have benefits [15]. Thus, the widespread implementation of laboratory tests determining DOAC drug levels will support care in an increasing proportion of the general population.

However, laboratory tests determining DOAC drug levels have not yet been implemented in all laboratories and are not available in many primary and secondary care institutions [3,16]. What are the barriers of implementation? Routine coagulation tests such as prothrombin time or activated partial thromboplastin time are not sensitive enough for the detection of DOAC [17,18,19,20]. A separate anti-Xa assay must be kept on hand for each of rivaroxaban, apixaban, and edoxaban, making it time-consuming and expensive at laboratories [3,16,17]. In this situation, prothrombinase-induced clotting time (PiCT) could demonstrate its advantages. The PiCT is a simple functional assay determining the clotting time after adding prothrombinase complex to the patient’s plasma [21]. Determination of PiCT is inexpensive and can be done rapidly [21]. It can be done on any coagulation device, and it can also be utilized for monitoring unfractionated heparin [22]. The accuracy of PiCT for the monitoring of unfractionated heparin was confirmed in several independent studies [21,22,23,24,25]. As long as essential similarities exist in the mechanism of action of unfractionated heparin and direct oral anti-Xa inhibitors, we hypothesized that PiCT would accurately determine the drug concentrations of rivaroxaban, apixaban, and edoxaban. We further hypothesized that PiCT would adequately predict clinically relevant drug concentrations, to decide upon reversal agents, scheduling of operations, and decisions for or against thrombolysis.

As part of a prospective cross-sectional study conducted in real-life clinical practice, we aimed to determine the accuracy of PiCT for the measurement of rivaroxaban, apixaban, and edoxaban drug concentrations. In addition, we aimed to assess whether clinically relevant drug levels could be predicted correctly.

## 2. Materials and Methods

### 2.1. Design, Setting, and Population

The present analysis was done as part of the Simple-Xa cross-sectional study [15,26,27]. Nine-hundred and thirty-two patients treated with rivaroxaban, edoxaban, or apixaban in clinical practice were included in 9 Swiss tertiary hospitals, between 2018 and 2019. We applied the following inclusion criteria: (1) use of rivaroxaban, apixaban, or edoxaban; (2) age above 18 years; (3) DOAC level ordered by the attending physician; and (4) signed general informed consent, if requested by the local authorities. Patients were excluded in case of (1) refused informed consent, (2) heparin use, (3) pre-analytical problems, (4) several DOAC taken, and (5) insufficient sample material. The blood samples were taken regardless of the time of last drug intake, to include the full range of drug concentrations. As a reference (gold) standard, ultra-high performance liquid chromatography-tandem mass spectrometry (LC-MS/MS) was performed on all samples. The detailed study design is illustrated in Figure 1. The study protocol was approved by the appropriate ethical committees. The study was conducted in accordance with the declaration of Helsinki.

### 2.2. Data Collection and Samples

Protocols detailing blood drawing, handling of samples, and transport are implemented at all institutions, ensuring appropriate pre-analytic conditions [28]. Venous blood was taken (tubes containing 1 mL trisodium citrate (0.106 mol/L) per 9 mL of blood; Monovette^®^, Sarstedt, Nümbrecht, Germany). After centrifugation using an appropriate scheme [29], aliquots were immediately frozen (stored at −80 °C until shipment). Shipment was done on dry ice, with a delivery time of 3–4 h. The samples were stored uninterrupted at −80 °C. All laboratory test results were exported automatically, to avoid typing mistakes. All data were stored in a secured and encrypted RedCAP database. Age, sex, and drug used were recorded.

### 2.3. Determination of the Prothrombinase-Induced Clotting Time

PiCT was determined using a Pefakit PiCT (DSM Pentapharm, Basel, Switzerland), using an Atellica COAG 360 device (Siemens Healthineers, Marburg, Germany). The assay has been described in detail elsewhere; we strictly followed the manufacturer’s instructions [21,22,24]. Citrated plasma samples were thawed in a water bath for 15 min at 37 °C. The measurements were made immediately after thawing. Of the sample, 50 µL was incubated for 180 s with an activator containing Russel’s viper venom, phospholipids, and activated factor Xa. Calcium chloride was added, and the clotting time was measured (s). Internal quality control was checked before and after each test run. Day-to-day and within-run imprecision was determined by 15 measurements of two samples (high and low DOAC concentrations).

### 2.4. LC-MS/MS Measurements

Rivaroxaban, Edoxaban, Edoxaban M4, and Apixaban plasma concentrations were determined using LC-MS/MS, as previously described [27]. The following substances were added to the plasma samples for protein precipitation and extraction: extraction buffer (MassTox TDM Series A, Chromsystems, Gräfelfing, Germany), acetonitrile:water 1:1 (*v*/*v*), and precipitation reagent containing the isotope labeled standards 13C6 rivaroxaban, 13CD3 apixaban, and 13CD2 edoxaban (MassTox TDM Series, Chromsystems, Gräfelfing, Germany). The samples were swirled and centrifuged, and water:methanol 8:2 (*v*/*v*) was added to the supernatant. Reversed-phase chromatography was used on a triple quadrupole mass spectrometer (Xevo TQ-S, Waters, Milford, CT, USA) coupled to a UPLC Acquity I-Class system (Waters, Milford, CT, USA) for analysis. The Edoxaban M4 and Edoxaban concentrations were summed up for the analysis.

### 2.5. Statistical Analysis

The study population was described using numbers/percentages or median/interquartile range. To determine the within-run and day-to-day imprecision in 15 measurements of two samples, with low and high drug concentrations, the coefficient of variation (SD/mean) was calculated. The Spearman correlation coefficient was used to determine the level of agreement between PiCT and LC-MS/MS measurements. Deming regression analysis was additionally calculated. Receiver–operating characteristics (ROC) curves were plotted to determine the thresholds of the PiCT. Sensitivities and specificities were calculated with respect to clinically relevant drug concentrations (30, 50, and 100 µg L^−1^) [9]. A power analysis was conducted, as described previously [27]. The statistical software Stata IC, version 16.1, was used for calculations (Stata Corp LP, College Station, TX, USA). The figures were drawn using Prism 8 (GraphPad Software, Inc., La Jolla, CA, USA).

## 3. Results

### 3.1. Patient Characteristics

Out of 932 patients included in the nine study centers, 851 were considered in the current analysis; a flow sheet is given in Figure 1. Thirty-five individuals were excluded due to additional use of heparin, two individuals because more than one DOAC was identified, five samples because of preanalytical issues, and thirty-nine samples because insufficient material was available. Four hundred and fourteen patients were treated with apixaban, 373 with rivaroxaban, and 64 with edoxaban. The median age was 76 (IQR 66 to 83), and 360 patients were female (42.3%). Patient characteristics are given in Table 1.

### 3.2. Association between PiCT Measurements and DOAC Concentrations

Figure 2 illustrates the relation between the PiCT results and rivaroxaban, edoxaban, and apixaban concentrations, details are given in Table 2. The overall correlation coefficient (r_s_) was 0.75 (95% confidence interval, CI 0.72, 0.78). In patients treated with rivaroxaban, r_s_ was 0.85 (95% CI 0.82, 0.88). It was 0.66 in the case of apixaban (95% CI 0.60, 0.71) and 0.78 with edoxaban (95% CI 0.65, 0.86). The slope of the regression equation was 0.15 for rivaroxaban, 0.12 for apixaban, and 0.10 for edoxaban (Table 2). The Y-intercept was 36.1 for rivaroxaban, 38.9 for apixaban, and 41.3 for edoxaban (Table 2).

### 3.3. Prediction of Clinically Significant Drug Levels

The distribution of PiCT results in patients with and without clinically relevant drug concentrations (30 µg L^−1^; 50 µg L^−1^; 100 µg L^−1^) is shown in Figure 3. Optimal cut-off values were derived using receiver–operating characteristics (ROC) curves: 36.0 s for 30 µg L^−1^, 39.6 s for 50 µg L^−1^, and 46.1 s for 100 µg L^−1^. The area under the ROC curve was 0.87, 0.89, and 0.90, respectively. The sensitivity in detecting a clinically relevant drug concentration was 85.1% in the case of 30 µg L^−1^ (95% CI 82.0, 87.7), 85.7% for 50 µg L^−1^ (95% CI 82.4, 88.4), and 85.1% for 100 µg L^−1^ (95% CI 80.9, 88.4). The specificities were 77.9% in the case of 30 µg L^−1^ (95% CI 72.1, 82.7), 77.3% for 50 µg L^−1^ (95% CI 72.5, 81.5), and 73.2% for 100 µg L^−1^ (95% CI 69.1, 76.9), respectively. The ROC curves are given in Figure 4.

## 4. Discussion

Using samples from 851 patients included in a large prospective cross-sectional study, we assessed the association between PiCT measurements and rivaroxaban, apixaban, and edoxaban drug concentrations, as measured with LC-MS/MS. The correlation was fair, both overall and in the case of the individual drugs. The highest correlation between PiCT times and LC-MS/MS measurements was observed in patients treated with rivaroxaban (r_s_ = 0.85). The sensitivity to detect clinically relevant drug levels was adequate, and it was correctly predicted in the majority of patients.

This is the first study assessing the association between PiCT measurements and DOAC drug levels in patient samples. Several previous studies observed the accuracy of PiCT for the monitoring of heparin. Brisset and colleagues assessed samples of 377 patients treated with unfractionated heparin [22]. A higher correlation with UFH concentrations than aPTT was found. Similarly, Buerki et al. found a high correlation between PiCT measurements and UFH concentrations in samples of 254 consecutive patients [21]. Raivio and colleagues observed PiCT measurements in 100 consecutive patients undergoing cardiovascular surgery and concluded that PiCT might be an alternative to anti-Xa assays in this setting [30]. Similar results were observed in other studies [23].

The strength of our study is that we observed a large number of patients in clinical practice, included from nine study centers. Furthermore, all three Xa-inhibitors were covered. As the study design reflects routine clinical practice, the results are straightforwardly applicable. In addition, LC-MS/MS measurements were taken, to apply an accurate reference standard. Our study has limitations as well. First, the number of patients taking edoxaban was limited, resulting in a restricted precision in edoxaban patients. Second, even though we sought to exclude all samples containing substances interfering with the PiCT assay, we cannot completely rule out the possibility that this may still have been the case for a few samples. Thus, unknown interferences may have affected the PiCT results on a case-by-case base. This may have contributed to the limited correlation with LC-MS/MS measurements. Third, we applied one protocol for PiCT measurements, rather than optimized protocols for each drug. Preliminary data suggested that this may increase accuracy. Fourth, the present study was not planned as a comprehensive pharmacokinetics study providing sufficient arguments for therapeutic drug management. Fifth, we did not assess different lots of the reagent, and we did not assess PiCT on different analyzers. Even though the manufacturer reports very low variability between batches, the user is recommended to check the variability between batches and analyzers using calibration plasmas.

We determined that the association between PiCT measurements and rivaroxaban, apixaban, and edoxaban drug concentrations was fair, and the majority of clinically relevant drug concentrations were predicted correctly. Our data suggest the application of PiCT in cases of urgent surgery, major bleeding, and planned thrombolysis. In the case of PiCT results above the thresholds, clinically significant drug levels can be assumed and decisions regarding reversal agents, scheduling of urgent surgery, and thrombolysis made. However, anti-Xa assays are available in many institutions, providing more accurate information [27]. However, there are conceivable situations in which screening for clinically relevant drug levels using PiCT makes sense. As an example, some laboratories provide PiCT in a 24/7 service for the monitoring of unfractionated heparin and do not want to implement yet another test. In this situation, we recommend using the cut-off values given in Figure 3. Future studies shall confirm our observations in other settings and patient populations.

## 5. Conclusions

In a large prospective study conducted in clinical practice, we assessed whether PiCT can accurately determine rivaroxaban, apixaban, and edoxaban drug concentrations and correctly predict clinically relevant drug levels. We found that the association between PiCT and LC-MS/MS measurements was fair, and the majority of patients with clinically relevant drug levels were predicted correctly. Even though anti-Xa assays provide more accurate information, PiCT might play a role in special settings, to estimate clinically relevant drug concentrations.

## Figures and Tables

**Figure 1 life-12-01027-f001:**
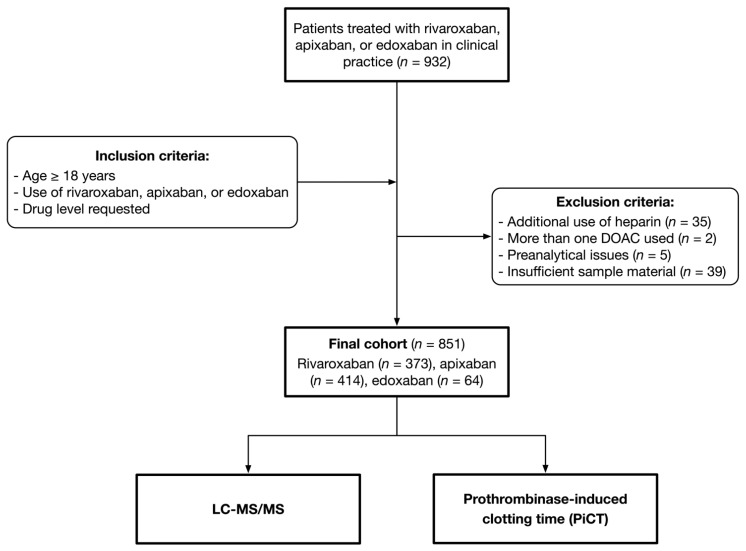
Patient flow diagram. We conducted a cross-sectional study, to assess the accuracy of prothrombinase-induced clotting time to measure rivaroxaban, apixaban, and edoxaban concentrations.

**Figure 2 life-12-01027-f002:**
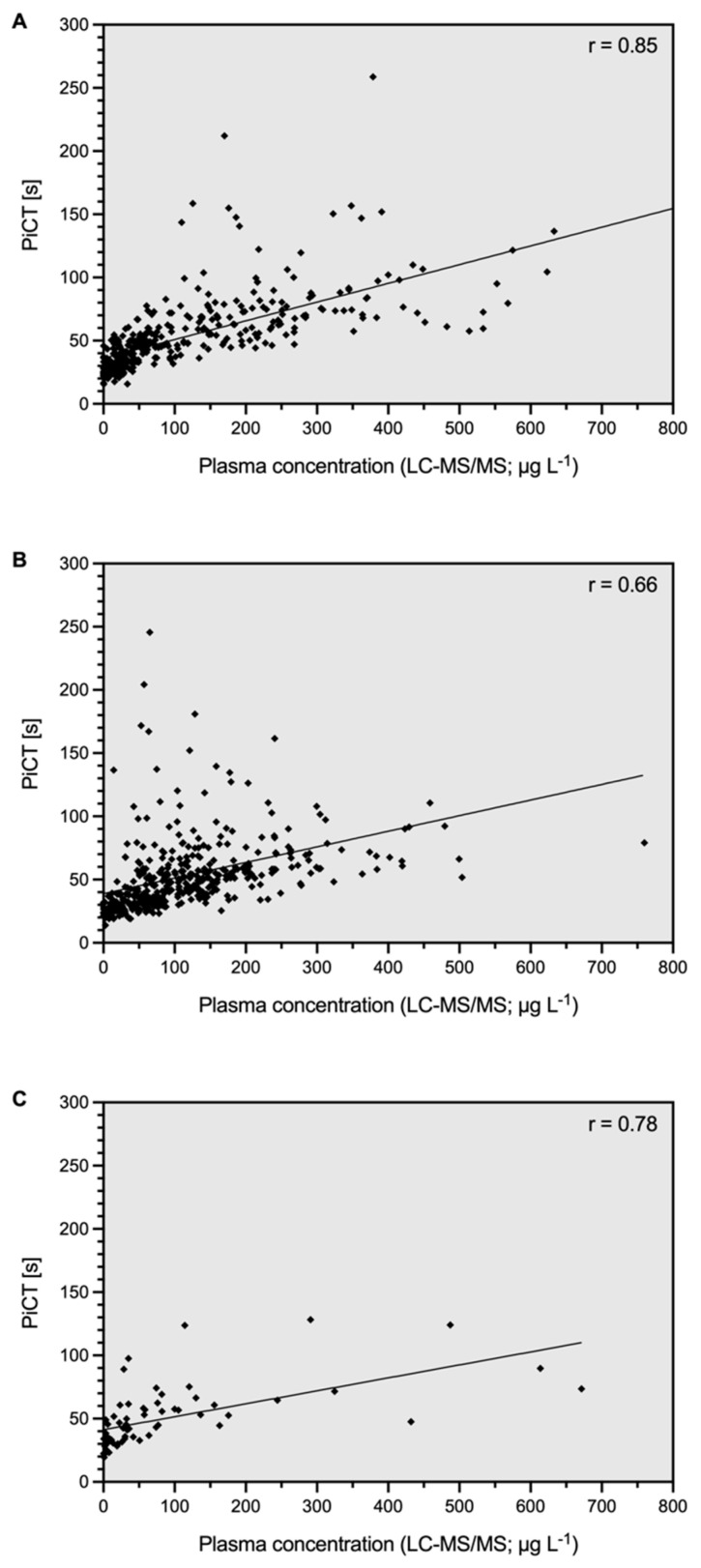
Association of prothrombinase-induced clotting time and (**A**) rivaroxaban, (**B**) apixaban, and (**C**) edoxaban drug concentrations. A cross-sectional study including patients from clinical practice was conducted (*n* = 851). Drug concentrations were determined using ultra-high performance liquid chromatography-tandem mass spectrometry (LC-MS/MS). The overall Spearman’s correlation coefficient (r_s_) was 0.75 (95% confidence interval 0.72, 0.78).

**Figure 3 life-12-01027-f003:**
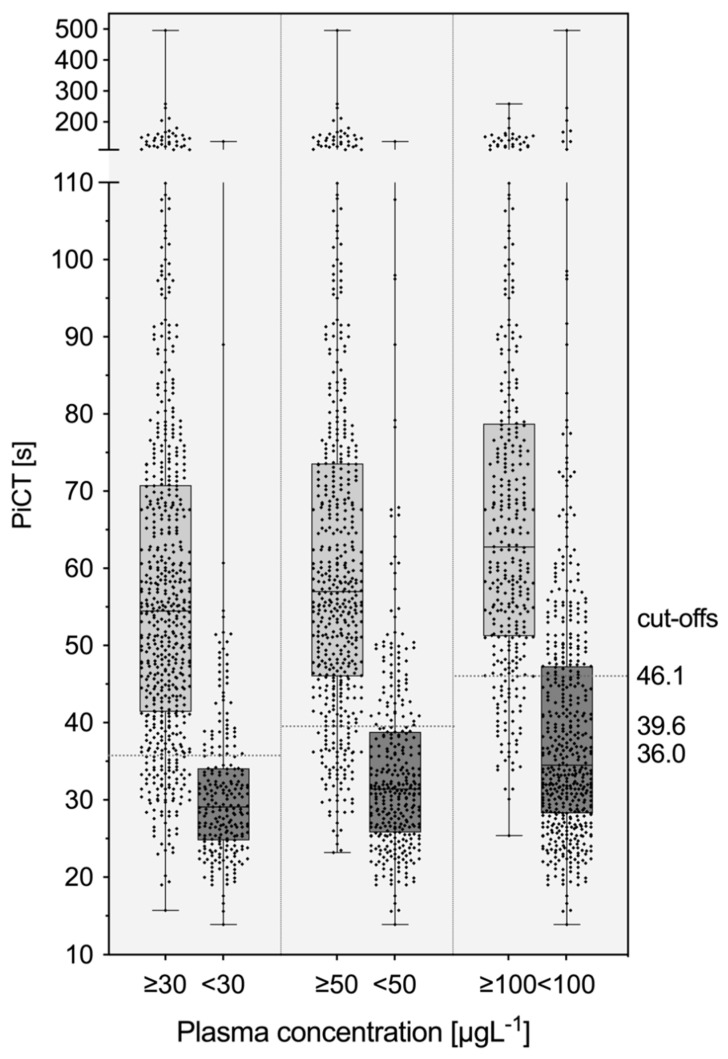
Distribution of prothrombinase-induced clotting times in patients with and without clinically relevant concentrations of rivaroxaban, edoxaban, and apixaban. Ultra-high performance liquid chromatography-tandem mass spectrometry (LC-MS/MS) was used to determine drug concentrations. Optimal thresholds are given at 36.0 s, 39.6 s, and 46.1 s, respectively.

**Figure 4 life-12-01027-f004:**
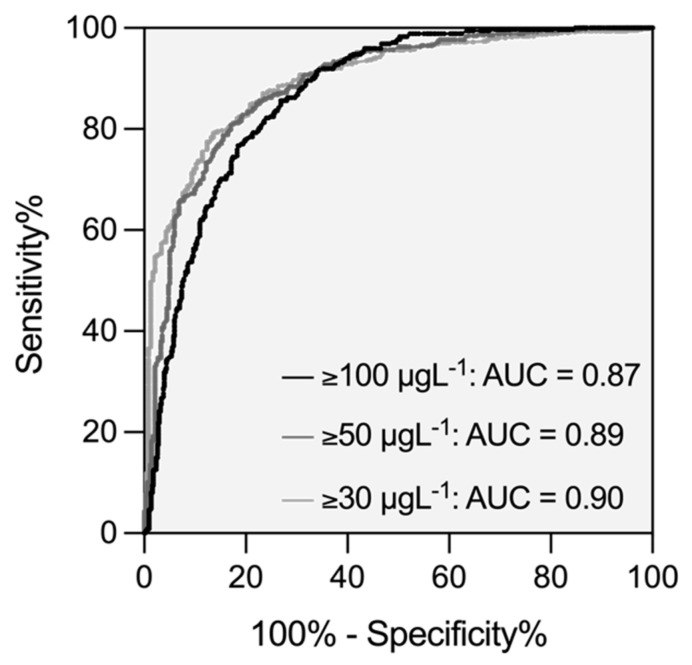
Receiver operating characteristics (ROC) curve of prothrombinase-induced clotting time clinically relevant concentrations of rivaroxaban, edoxaban, and apixaban (30 µg L^−1^; 50 µg L^−1^; 100 µg L^−1^). The sensitivity was 85.1% for 30 µg L^−1^ (95% CI 82.0, 87.7), 85.7% for 50 µg L^−1^ (82.4, 88.4), and 85.1% for 100 µg L^−1^ (80.9, 88.4). Specificities were 77.9 % in the case of 30 µg L^−1^ (95% CI 72.1, 82.7), 77.3 % for 50 µg L^−1^ (95% CI 72.5, 81.5), and 73.2% for 100 µg L^−1^ (95% CI 69.1, 76.9), respectively.

**Table 1 life-12-01027-t001:** Patient characteristics. In a multicenter cross-sectional study, patients taking rivaroxaban, edoxaban, or apixaban in clinical practice (*n* = 851) were included. N, number. IQR, interquartile range.

	Patients Treated With			
	Any Drug	Rivaroxaban	Apixaban	Edoxaban
Patients (*n/%*)	851 (100)	373 (43.8)	414 (48.7)	64 (7.5)
Age (*median/IQR*)	76 (66, 83)	74 (63, 83)	78 (68, 82)	75 (56, 81)
Female sex (*n/%*)	360 (42.7)	164 (44.1)	167 (40.9)	29 (46.0)

**Table 2 life-12-01027-t002:** Accuracy of the prothrombinase-induced clotting time (PiCT) in measuring rivaroxaban, edoxaban, and apixaban drug concentrations. Eight-hundred and fifty-one patients were included. DOAC drug levels were measured using ultra-performance liquid chromatography-tandem mass spectrometry (LC-MS/MS). Spearman’s correlation coefficient (r_s_) and coefficients of Deming regression are given.

	Patients Treated with			
	Any Drug	Rivaroxaban	Apixaban	Edoxaban
Spearman’s correlation coefficient (*95% CI*)	0.75 (0.72, 0.78)	0.85 (0.82, 0.88)	0.66 (0.60, 0.71)	0.78 (0.65, 0.86)
Deming regression Slope (*95% CI*)	0.14 (0.12, 0.16)	0.15 (0.12, 0.18)	0.12 (0.09, 0.16)	0.10 (0.04, 0.17)
Y-intercept (*95% CI*)	37.5 (35.2, 39.9)	36.1 (33.4, 38.8)	38.9 (34.5, 43.2)	41.3 (35.6, 47.0)

## Data Availability

The raw data supporting the conclusions of this article will be made available by the authors, without undue reservation.

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
