# Peer review of "Prothrombinase-Induced Clotting Time to Measure Drug Concentrations of Rivaroxaban, Apixaban, and Edoxaban in Clinical Practice: A Cross-Sectional Study"

_life, 2022, doi:10.3390/life12071027_

Round 1

Reviewer 1 Report

The authors present a cross-sectional study that aims to quantify DOACs concentration correlating it with the PiCT measurement during therapeutic drug monitoring. The aim of this work is to determine if PiCT can be applied as a clinical pharmacokinetics test for therapeutic drug monitoring for dose adjustments or other clinical scenarios for DOACs. The work is very interesting, within the context of the SI and generally well presented. However, I believe it would greatly benefit if the authors could elaborate some more on important details, especially in their introduction and methods section. For example what are the general pharmacokinetic profiles for DOAcs? What are the target values regarding their pharmacodynamics. Moreover, in the results, how uniform is the cohort regarding treatment duration etc. How many samples over each time point etc. 

Some additional comments: 

1) Does low strength aspirin interferes with the PiCT? Why not included in the exclusion criteria or any other drugs (NSAIDs, Calcium supplements etc.) that could have a PK/PD drug interaction effect? The authors state the DDIs factor in discussion. But since they record the co-medications this could be easily determined. 

2) Usually, for anticoagulants such as warfarin, there is a clinical guidance for +/- 2-3 days gap from last dose to operation in order the clotting mechanism to be restored. Is it considered in the correlation?

3) "The blood samples were taken regardless of the time of last drug intake to include the full range of drug concentrations". Yes but how many were given in each time point? Is it in Cav,ss for all subjects? Are there samples before the next dose or after it? Are the correlations take into account the time point, the steady-state or other PK parameters?

4) line 145. If possible use only numbers or words to state numbers (nine) 

5) What was the dose for each drug? It could be placed in the table 1. 

6) It would be interesting to see the Pharmacokinetic parameters from the LC/MS analysis.  

7) Figure 2 Could be split in 3 sub plots so we can better observe the correlation with the CIs lines.  

8) Did the authors considered for their correlation to adjust LC/MS/MS concentrations with body weight or dose etc.?

9) How the concentrations 30 µgL-1,; 50 µgL-1 ; 100 µgL-1 are considered clinically relevant? Are they referred to all 3 drugs? It is little bit confusing.

10) It is also confusing how the cut-offs were extracted from ROCs. These cut offs refer to which drug? Regardless of the dose or time of administration?

11) Please fix the ROC legend L-1 to upper case

12) The study would greatly benefit if it could show a clinical scenario that it is applied. 

13) Line 222 "Results are straightforwardly applicable to routine clinical practice". The correlation of apixaban is 0.66. The authors debate it as a limitation. So how can this statement be supported? Usually we expect a correlation R2 >0.9 when we compare it with an old method. Although they state there are anti-Xa assays that provide more accurate information we do not see any comparison. 

14) A supplementary file with chromatograms, C-t graphs or additional information would be useful. 

Author Response

On behalf of all co-authors, we would like to thank you for taking the time to carefully review our manuscript. We feel that your comments have helped to a substantially improve the quality of the manuscript. Point-by-point responses are listed below. We hope that we answered all issues to your satisfaction and that our manuscript is now acceptable for publication in Life.

As a response to your general comments, we would like to clarify that we did not intend to conduct a comprehensive pharmaco­kinetics study serving as a base for therapeutic drug monitoring. No such application has been established, and target values do not exist. DOAC drug concentrations are recommended measuring in case of (a) massive bleeding, (b) urgent surgery, and (d) planned thrombolysis supporting clinical decisions regarding reversal agents, scheduling operations, and decisions for or against thrombolysis. The recommended thresholds for clinical decisions are 30 µg/L (high bleeding risk or life-threatening bleeding), 50 µg/L (normal bleeding risk), and 100 µg/L (planned thrombolysis). In our study, we aimed to assess the diagnostic accuracy of PiCT in this situation. To do this as accurately as possible, we designed a study cohort that resembles the target population of the test closely. Thus, we included a large cohort of unselected patients (n= 851) with DOAC drug levels requested (inclusion criterion). This was done irrespectively of the time point of drug intake (because in clinical practice it is not known in this clinical situation). Each patient was only tested once. We found that the association between PiCT and LC-MS/MS measurements was fair, and the majority of patients with clinically relevant drug levels were predicted correctly. Thus, we were able to answer this research question. We added several explanations and refer to previous statements to make this clear (introduction, line 54 to line 57 and line 78 to line 80; discussion, line 248 to line 264; line 270 to line 273; conclusions, line 286 to line 288).

  1. Thank you for the possibility to clarify this point. Heparin and other anticoagulants are known to affect coagulation tests like PT or PiCT. In contrast, it is well established that aspirin, NSAID, and similar drugs do not affect coagulation tests. Thus, we did not record the presence of these drugs (e.g. doi.org/10.4065/82.7.864).
  2. We fully agree with the reviewer, guidelines recommend intervals between last DOAC intake and operations (e.g. doi:10.1111/jth.13227). However, the measurement of drug levels is recommended in case of (a) urgent surgery (can an operation be performed now?), (b) massive bleeding (must a reversal agent be given?), (c) planned thrombolysis (can thrombolysis be performed safely?) (e.g. doi:10.1111/jth.13227; doi: 10.4414/smw.2018.14598). Often, the time of the last dose is unclear in clinical practice, and the exact drug concentration cannot be accurately estimated due to a considerable variability between different patients (e.g. DOI: 10.1111/jth.13747). Please see response to general comments above.
  3. Thank you for raising this issue. We designed a cohort that best matches the scientific question mentioned above. Thus, it was done irrespectively of the time point of drug intake (because it is not known in this clinical situation). Also, the drugs taken were not recorded. PK parameters of the individual patients are not known. Please see response to general comments above.
  4. Thank you. We changed the manuscript accordingly at line 154.
  5. Thank you for raising this issue. Because of the design of the study was the dose not recorded. Please see discussion above.
  6. Thank you for the possibility to clarify this point. Each patient was only tested once and PK parameters were not obtained. Please see discussion above.
  7. We fully agree with the reviewer and changed Figure 2 accordingly.
  8. Thank you for raising this issue. As long as the endpoint of the study was to assess the accuracy regarding clinically relevant drug concentrations (which were measured by LC-MS/MS), no adjustments with body weight, dose etc. were done. Please see discussion above.
  9. Thank you. The justification of the cut-off values is given above; the cut-off values apply to all three drugs indeed (e.g. DOI: 10.1111/jth.13227).
  10. Thank you. The aim was to assess the utility of the PiCT as a universal assay for all three drugs. Thus, the thresholds (of the PiCT assay) were derived from the ROC curve regarding the clinically relevant cut-off values (DOAC concentrations) as defined current guidelines. Regardless of dose or time of administration.
  11. Thank you for spotting. We changed the legend accordingly (line 213 to 218).
  12. Please see reply to general comments above.
  13. Thank you for raising this issue. This sentence refers to the “applicability” in terms of the QUADAS criteria (DOI: 10.7326/0003-4819-155-8-201110180-00009). The study population perfectly resembles the target population of the test. We reworded this sentence to make this clear (line 239 to line 240).
  14. Unfortunately, no chromatograms, C-t graphs or similar information was recorded.

Reviewer 2 Report

Overall the data shows the potential utility of PiCT to assess the presence of various anti-Xa DOACs, but on a case by case basis there is a chance to overcall the presence of DOACs due to unknown confounders or interferences. Thus, the manuscript should have additional caveats. Also, there was no evaluation of lot to lot variation in the PiCT assay, and it is unclear if the values noted (in sec) will actually hold true for all lots of the reagent on all instruments. I have the following comments suggestions:

1. Add to the statement of limitations some additional caveats, including (or similar): (a) Overall the data shows the potential utility of PiCT to assess the presence of various anti-Xa DOACs, but on a case by case basis there is a chance to overcall the presence of these DOACs due to unknown confounders or interferences. (b) We did not assess different lots of PiCT reagent, and so the times in sec may vary according to the reagent lot. It is recommended that users check each lot with available calibration plasmas to check lot to lot variation. (c) We did not assess the PiCT assay on additional instruments, and thus some variation of PiCT times based on instrument clot detection procedures may be expected.

2. Did the authors further investigate the potential reasons for the major sample discrepancies. For example, concomitant heparin, presence of lupus anticoagulant, factor deficiencies, all of which may affect PiCT? Many of these do not appear to have been identified as exclusion criteria in Fig 1.

3. Line 211: "This is the first study assessing the association between PiCT measurements and DOAC drug levels in patient samples." Not sure this is true. A PubMed search of (Prothrombinase-induced clotting time) AND (DOAC OR rivaroxaban OR apixaban OR edoxaban) identifies 12 papers; authors should discuss their findings against any relevant papers.

4.  Occasional use of comma separator instead of decimal points (eg: lines 187-189" was 85,1 % in the case of 30 μgL-1 (95 % CI 82.0, 87.7), 85,7 % in 50 μgL-1 (95 % CI 82.4, 187 88.4), and 85,1 % in 100 μgL-1 (95 % CI 80.9, 88.4). The specificities were 77.9 % in the case 188 of 30 μgL-1 (95 % CI 72.1, 82.7), 77,3 %...

5. Line 101: "Venous blood was taken (tubes containing 1 mL trisodium citrate per mL of blood)." must be in error, as citrate is usually used at 10% by volume, so probably containing 0.1 mL trisodium citrate per mL of blood.

6. line 67: "...are not sensitive enough for the detection of detect DOAC"

Author Response

On behalf of all co-authors, we would like to thank you for taking the time to carefully review our manuscript. We feel that your comments have helped to a substantially improve the quality of the manuscript. Point-by-point responses are listed below. We hope that we answered all issues to your satisfaction and that our manuscript is now acceptable for publication in Life.

1) We fully agree with the reviewer and added similar sentences at various positions in the limitation’s sections of the manuscript (line 243 to line 258).

2) Thank you for the possibility to clarify this point. We excluded patients with concomitant heparin treatment, more than one DOAC found, and preanalytical issues in the LC-MS/MS measurements (please see line 92 to line 94 and Figure 1). Even though we visually checked the dataset in the data cleaning process for unexplained long PT or APTT results (which were not found), this was not formally defined as an exclusion criterion. We expanded the current statement in the discussion section to clarify this limitation (line 240 to line 244).

3.) Thank you for raising this issue. After checking again, we can confirm that most previous studies used spiked samples and the only remaining study did not observe the association of PiCT with DOAC drug concentrations (line 266 to line 227).

4.) Thank you. We have corrected all those typos (various positions in the manuscript).

5.) Thank you for you spotting this typing error. We rephrased this sentence at line 108 to line 110.

6.) Thank you for spotting. We corrected this typing error at line 67.

Round 2

Reviewer 1 Report

The authors provided sufficient answers to the initial comments. I believe the manuscript can be further processed for publication. I would recommend figure 2 and figure 3 to be enhanced  in size and resolution in the final version. 

Author Response

We fully agree with the reviewer. We replaced figures 2 and 3 with figures enhanced in size and resolution.